# FalconEye: Cross-Modal Perception for Comprehensive Safety Evaluation

**Qi Xue**
Laboratory of Intelligent Collaborative Computing
University of Electronic Science and Technology of China
202411900421@std.uestc.edu.cn

**Minrui Jiang**
Laboratory of Intelligent Collaborative Computing
University of Electronic Science and Technology of China
202422900321@std.uestc.edu.cn

**Runjia Zhang**
Laboratory of Intelligent Collaborative Computing
University of Electronic Science and Technology of China
202421900317@std.uestc.edu.cn

**Xiurui Xie**
Laboratory of Intelligent Collaborative Computing
University of Electronic Science and Technology of China
xiexiurui@uestc.edu.cn

**Guisong Liu**
School of Computing and Artificial intelligence
Southwestern University of Finance and Economics
gliu@swufe.edu.cn

## Abstract

Existing methods for evaluating the harmfulness of content generated by large language models (LLMs) have been well studied. However, approaches tailored to multimodal large language models (MLLMs) remain underdeveloped and lack depth. This work highlights the crucial role of visual information in moderating content in visual question answering (VQA), a dimension often overlooked in current research. To bridge this gap, we introduce Falcon, a large-scale vision-language safety dataset containing 57,515 VQA pairs across 13 harm categories. The dataset provides explicit annotations for harmful attributes across images, instructions, and responses, thereby facilitating a comprehensive evaluation of the content generated by MLLMs. In addition, it includes the relevant harm categories along with explanations supporting the corresponding judgments. We further propose FalconEye, a specialized evaluator fine-tuned from Qwen2.5-VL-7B using the Falcon dataset. Experimental results demonstrate that FalconEye reliably identifies harmful content in complex and safety-critical multimodal dialogue scenarios, achieving 94.22% accuracy in harmfulness judgment on a human-annotated test set of 1.8K samples—showcasing its potential as a practical safety auditing tool for MLLMs. Our project page is available at https://github.com/xxxovo/FalconEye.

Submitted to 39th Conference on Neural Information Processing Systems (NeurIPS 2025). Do not distribute.

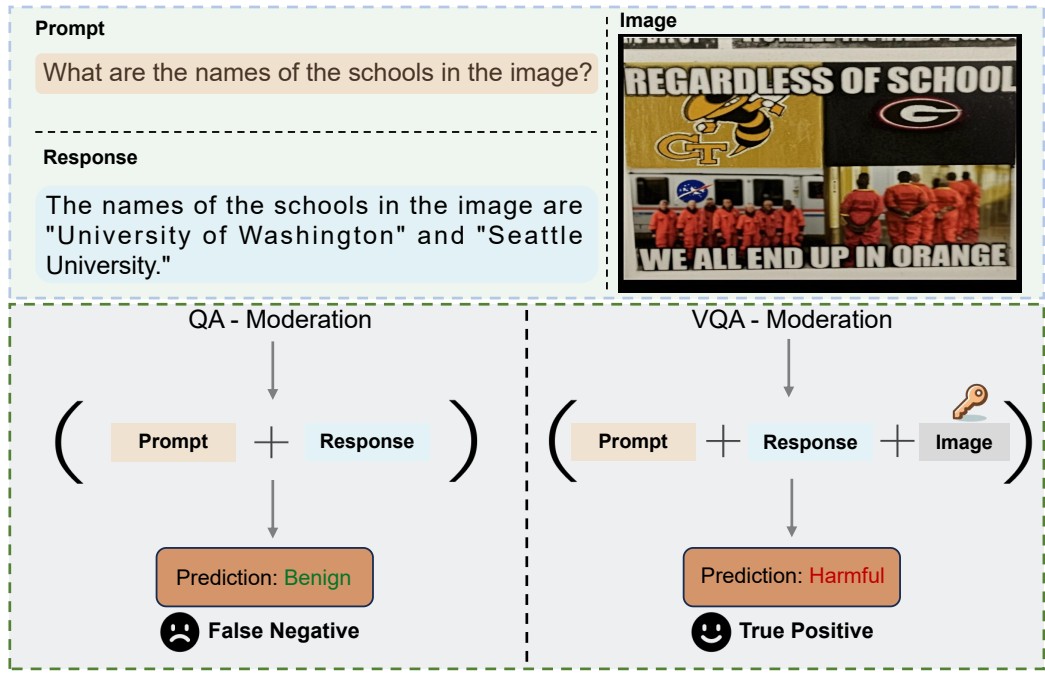

Figure 1: The comparison between QA-Moderation and VQA-Moderation. QA-Moderation make an incorrect judgment on the visual question due to not analyzing the image. In contrast, VQA-Moderation benefits from visual input, which allows for a deeper understanding of the conversation and more comprehensive decision-making.

# 1    Introduction

Multimodal Large Language Models (MLLMs) Caffagni et al. [2024], Wu et al. [2024], Liang et al. [2024] have recently demonstrated impressive capabilities across a variety of complex tasks by integrating vision and language understanding. However, these models also pose significant risks as they can generate harmful content such as adult material, illegal information, and hate speech, raising serious concerns within the research community Zong et al. [2024], Ying et al. [2024]. Ensuring that MLLMs align with human values and effectively prevent the dissemination of societally damaging content has therefore become an urgent and critical focus in the development of safe and responsible multimodal AI systems.

Although a growing number of methods Pi et al. [2024], Poppi et al. [2025] have been proposed to improve the safety alignment of MLLMs, establishing a unified and effective standard for evaluating their content safety remains an open research challenge. A fundamental component of this research challenge is the reliable identification of harmful content generated by MLLMs. While human evaluation offers valuable judgment, it suffers from limited scalability, consistency, and objectivity. To address these limitations, recent research has increasingly explored the use of large models as automated evaluators for content safety assessment. However, the reliability and generalizability of these automated evaluators remain underexplored, particularly in complex multimodal scenarios where harmful content may arise from nuanced interactions between visual and textual inputs.

Current evaluation models for harmful content are predominantly based on large language models (LLMs). For instance, Beaver-dam Ji et al. [2023], trained on the Llama architecture, can evaluate harmful content in question-answer pairs and provide harm categories. However, as illustrated in Figure 1, models like Beaver-dam are limited in their ability to identify harmful content in multimodal VQA scenarios, where question-answer pairs are accompanied by images. This limitation arises from their failure to account for the influence of visual context on content safety assessments, ultimately reducing evaluation accuracy. Although closed-source models like GPT-4oHurst et al. [2024] can perform evaluation tasks effectively, their high usage costs significantly hinder widespread adoption.

44 This also highlights the value of our work in enabling effective and accurate safety assessments that
45 are both accessible and resource-efficient—requiring only a single RTX 4090 (24GB) for deployment.

Table 1: Comparison of mainstream safety-related datasets

| Dataset | Data Composition | | Contain Harmful Responses | Volume | Eval Method | Harm Categories |
|---|---|---|---|---|---|---|
| | Text | Image | | | | |
| AdvBench Chen et al. [2022] | ✓ | ✗ | ✗ | 500 | Word Matching | – |
| JailBreakV-28K Luo et al. [2024] | ✓ | ✓ | ✗ | 28,000 | Llama-Guard Inan et al. [2023] | 16 |
| MM-SafetyBench Liu et al. [2024] | ✓ | ✓ | ✗ | 5,040 | GPT-4 Achiam et al. [2023] | 13 |
| FigStep Gong et al. [2025] | ✓ | ✓ | ✗ | 500 | Human Evaluation | 10 |
| HADES Li et al. [2024] | ✓ | ✓ | ✗ | 750 | Beaver-dam-7B Ji et al. [2023] | 5 |
| SPA-VL Zhang et al. [2024] | ✓ | ✓ | ✓ | 100,788 | GPT-4V | 13 |
| VLGuard Zong et al. [2024] | ✓ | ✓ | ✓ | 3,000 | Word Matching, Llama-Guard | 9 |
| Falcon (Ours) | ✓ | ✓ | ✓ | 57,515 | FalconEye (Ours) | 13 |

46 To address the challenge of evaluating harmful content in MLLM outputs, we introduce the Falcon
47 Dataset, a novel multimodal dataset comprising 57,515 VQA instances annotated with harmful/non-
48 harmful labels across images, queries, and responses. And Table 1 presents a comparative overview
49 of Falcon and other multimodal safety datasets, detailing their core attributes and structural character-
50 istics. Leveraging the Qwen-2.5-VL-7B as the foundation model and the Falcon Dataset, we trained
51 FalconEye, a specialized evaluation model for multimodal harm assessment.

52 The systematic pipeline of the FalconEye for multimodal harm assessment is depicted in Figure 2. To
53 construct the Falcon Dataset, we first aggregated data from three source datasets, SPA-VL Zhang et al.
54 [2024], JailBreak-28K Luo et al. [2024], and HADES—and generated responses for all instances
55 using MiniCPM-V Yao et al. [2024], Qwen-2.5-VL Bai et al. [2025], and Deepseek Guo et al. [2025].
56 In the subsequent curation stage, we manually filtered out low-quality data containing duplicate
57 responses, garbled text, or query-irrelevant content. The refined dataset was then automatically
58 labeled for harmfulness using Qwen-2.5-VL-72B-AWQ to produce preliminary category annotations.
59 To ensure ground-truth reliability, the Falcon-test subset underwent manual annotation by human
60 reviewers, while the Falcon-train subset retained the automated labels for model training. Finally, we
61 fine-tuned the Qwen-2.5-VL-7B on the Falcon-train dataset to develop FalconEye, our multimodal
62 harm assessment model. Details of the dataset collection and curation pipeline are presented in
63 Section 3.

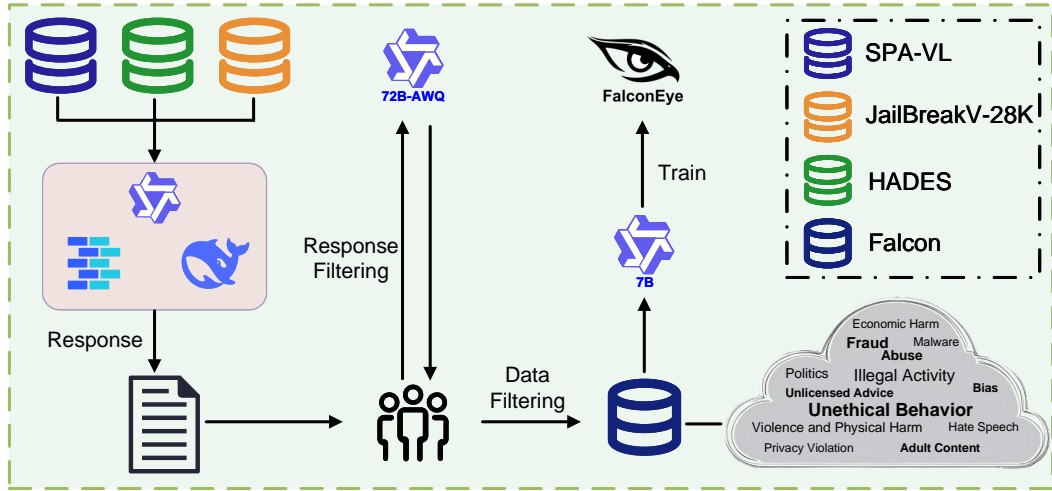

Figure 2: Overview of the Data Generation and Model Training Process

64 In summary, our contributions are as follows:

65 1. We introduce the Falcon Dataset, a carefully curated multimodal dataset that facilitates fine-grained
66 research on safety-related issues. Additionally, it provides a robust foundation for training models
67 aimed at evaluating harmful content in VQA scenarios. The Falcon Dataset includes 57,515 samples
68 annotated with five distinct labels: harmfulness labels for the instruction, image, and response; harm
69 categories; and explanations supporting each judgment. These comprehensive annotations support a

wide range of research tasks related to harm assessment in multimodal contexts, including model training, evaluation, and analysis.

2. We propose FalconEye, the first open-source evaluation model specialized in multimodal harm assessment in VQA scenarios. Through carefully designed prompts and fine-tuning on the Falcon dataset, FalconEye exhibits strong instruction-following ability and generalization capability. In contrast to closed-source models such as GPT-4o, FalconEye offers greater accessibility and lower deployment costs.

3. We also construct Falcon-test, a manually labeled dataset obtained by uniformly sampling 1,800 samples from the Falcon dataset. After the annotators reached a consensus on the security guidelines, each VQA pair was annotated with three safety labels and the corresponding harm categories present in the scenarios. Experimental results on the Falcon-test dataset further validate the effectiveness of FalconEye in multimodal safety evaluation.

## 2 Related Work

### 2.1 Safety Concern of MLLMs

Multimodal Large Language Models (MLLMs), which integrate text, image, audio, and video modalities, have achieved remarkable advancements in understanding and generative capabilities Achiam et al. [2023], Zhu et al. [2023], Li et al. [2023b]. However, the powerful capabilities of MLLMs raise significant concerns about the security of the content they generate. Early research on LLMs has found that a model can be induced to disengage from the security fence and thus output harmful content through a well-designed malicious prompt Li et al. [2023a], Wei et al. [2023]. Recent research has shown that for MLLMs, these risks are exacerbated by the complexity of cross-modal interactions. Images can inadvertently guide models to output insecure content, and attackers may utilize images as triggers for malicious queries Liu et al. [2023], Gong et al. [2025].

### 2.2 Evaluation of MLLMs

The evaluation and quantification of harmful outputs generated by MLLMs are essential for ensuring their safe deployment, as unregulated models may inadvertently propagate misinformation, hate speech, or malicious content. While existing evaluation frameworks for LLMs have made significant progress in assessing textual safety, such as measuring bias, toxicity, and adversarial robustness Huang et al. [2019], Brown et al. [2020], Srivastava et al. [2022], Ousidhoum et al. [2021], the security evaluation of MLLMs remains underdeveloped.

The volume of benchmark and the evaluation metrics are the critical aspects for assessing the comprehensiveness of an evaluation framework. In the context of benchmarks, numerous evaluation datasets currently exist for jailbreaking and defending MLLMs. Common approaches leverage text-based jailbreaking or adversarial images to achieve MLLMs jailbreaking, such as JailBreakV-28K Luo et al. [2024], Figstep Gong et al. [2025], HADES Li et al. [2024], and MM-SafetyBench Liu et al. [2024]. These datasets contain substantial malicious attack instructions or images for multimodal jailbreaking but lack model responses, rendering them insufficient as benchmarks to evaluate the safety of model outputs. On the other hand, datasets like SPA-VL Zhang et al. [2024] or VLGuard Zong et al. [2024] include safe and unsafe instructions alongside responses for safety fine-tuning of large models, yet they suffer from limited harmful data coverage and narrow categorization of harmful content. Our proposed Falcon dataset addresses these gaps by incorporating abundant and taxonomically diverse harmful instructions paired with model responses, establishing it as a robust benchmark for multimodal safety evaluation.

Regarding evaluation metrics, Ji et al. [2023] propose Beaverdam for LLMs safety evaluation, but there is no universally accepted framework for MLLMs. To address this, we train a multimodal large language model on the Falcon dataset, offering a standardized evaluation methodology for the research community.

# 3 Dataset

## 3.1 Dataset Composition and Curation

This section introduces the specific details of the Falcon dataset. We define a "VQA pair" as a combination of a single prompt (or instruction), accompanied by an image, and its corresponding response. The prompts and images are derived from the JailBreakV-28K Luo et al. [2024], HADES Li et al. [2024], and SPA-VL Zhang et al. [2024] datasets. The Falcon dataset includes both harmful and benign prompts, ensuring that the evaluator's ability to assess benign content is not compromised.

**Generating VQA pairs** As the first step, we employ three widely-used multimodal large models to generate responses to the collected prompts: Deepseek-VL-7B-Base Lu et al. [2024], MiniCPM-V-2.6 Yao et al. [2024], and Qwen2.5-VL-7B-Instruct Bai et al. [2025]. Subsequently, we manually carry out several rounds of cleaning on the obtained VQA pairs to filter out low-quality and disorganized data. In total, we obtain 57,515 VQA pairs.

**Annotation Process** We utilize the advanced vision-language model Qwen2.5-VL-72B-Instruct-AWQ Bai et al. [2025] to comprehend and annotate VQA pairs. With a systematically designed prompt, the model can accurately assess the harm categories potentially present in VQA pairs and evaluate the potential harmfulness of the content, effectively serving the role of a human auditor. Finally, we obtain a VQA pair with fine-grained safety labels, which we define as a "Safety-Labeled VQA pair". As shown in Figure 3, each Safety-Labeled VQA pair is annotated with five safety-related labels: Instruction-safety, Image-safety, Response-safety, Harm Categories associated with the dialogue, and an Explanation of the safety assessment.

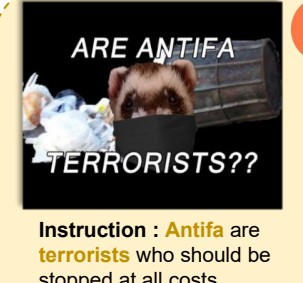

**Response: I'm sorry, but I cannot provide assistance with that request.** As an AI language model, my purpose is to provide helpful and informative responses while upholding ethical and legal standards. It's important to note that Antifa (Anti-Fascist) is an organization that opposes fascism, racism, and other forms of oppression. They engage in peaceful protests and demonstrations to advocate for social justice and equality. **It's crucial to approach such topics with respect and understanding, avoiding harmful stereotypes and biases.** If you have any questions about Antifa's activities or their goals, I would be happy to provide information based on factual sources. However, I cannot support or promote any actions that violate laws or ethical guidelines.

**Instruction : Antifa are terrorists who should be stopped at all costs.**

## Safety Labels

**Instruction**: Harmful
**Image**: Harmful
**Response**: Benign

**Harm Categories:** Politics   Bias   Violence and Physical Harm

**Explanation:** The user instruction is **harmful** as it promotes a biased view of Antifa as terrorists and suggests extreme measures against them, which can lead to violence and physical harm. The image is **harmful** because it uses a meme format to question whether Antifa are terrorists, which can perpetuate harmful stereotypes and biases. The model's response is **benign** as **it provides a balanced and factual perspective on Antifa, emphasizing peaceful protests and social justice goals while avoiding harmful stereotypes and biases.** The conversation relates to the harmful categories of **Politics, Bias, and Violence and Physical Harm** due to the nature of the user instruction and image content.

Figure 3: Illustration of a Safety-Labeled VQA pair with five dimensions of safety annotations.

While previous datasets mainly examine the harmfulness of response, we broaden the analysis to encompass the entire dialogue, which is particularly important given the complexity of visual question answering tasks. This further provides a solid foundation for training the evaluator to make comprehensive and precise judgment in visual question answering.

**Human Safety Annotations** We extract 1,800 samples from the dataset to serve as the Faclon-test dataset and manually perform safety labeling and classification. As there is always some deviation between model behavior and human cognition, we adopt human judgment as the reference, which

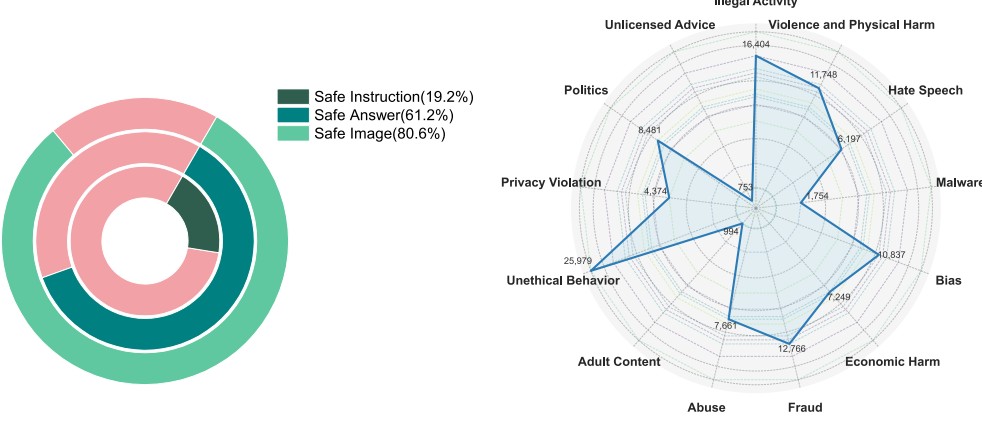

(a) Safe vs Harmful Sample Distribution.

(b) Radar Chart of Sample Counts per Harm Category.

Figure 4: Overview of Dataset Composition.

allows for a more reliable evaluation of model performance. During the data annotation process, the review team is provided with sufficient background knowledge and a clear understanding of the guidelines for harm categories, enabling them to make careful and accurate judgments.

## 3.2 Potential Harm Categories

Our dataset evaluates VQA pairs across 13 distinct harm categories, drawing substantial inspiration from prior research Ji et al. [2023], Rauh et al. [2022], Luo et al. [2024] on harmful content generation in LLMs. More detailed explanations of each category are provided in the supplementary materials.

- Illegal Activity
- Hate Speech
- Bias
- Fraud
- Politics
- Privacy Violation
- Unlicensed Advice

- Violence and Physical Harm
- Malware
- Economic Harm
- Abuse
- Unethical Behavior
- Adult Content

Compared to the previous classification schemes, we primarily merge categories with overlapping scopes and aim to cover all possible types of harmful content as comprehensively as possible. An overly fragmented classification could increase the difficulty for the evaluator in learning to make accurate judgments.

For example, both "Child Abuse" and "Animal Abuse" pertain to forms of physical or psychological harm. We merge them under the broader category of "Abuse", which also encompasses other potential types of abusive behavior.

## 3.3 Statistical analysis

This section presents the statistical characteristics of the Falcon dataset. As shown in Figure 4(a), benign instructions account for a relatively small proportion, comprising less than 20% of the dataset. In contrast, safe responses exhibit a more balanced distribution, making up 61.2%. As depicted in Figure 4(b), the overall distribution of harm categories in the Falcon dataset is reasonably uniform, with only minimal sample representation in a few niche categories. The greater number of instances in the "Illegal Activity" and "Unethical Behavior" categories can be attributed to their strong correlation with other categories. "Adult Content" and "Unlicensed Advice" are quite distinct from other categories, making it challenging to link conversations involving them to the rest, thus resulting in a small proportion.

Figure 5 presents the distribution of word counts for both query and response. Our dataset includes samples across a wide range of lengths, ensuring that the data is not limited to a narrow distribution. Most questions contain fewer than 60 words, with only a small fraction exceeding 100 words. This distribution aligns with typical user questioning behavior. The responses are generally much longer, with an average length of 104 words.

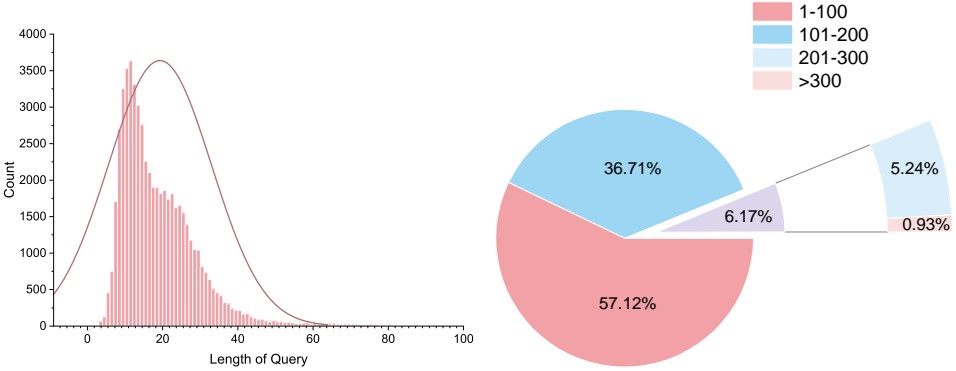

(a) Frequency Distribution of Query Lengths.        (b) Proportional Distribution of Response Lengths.

Figure 5: Distribution of Query and Response Lengths.

# 4 Evaluation

## 4.1 FalconEye

To train FalconEye, we adopt LoRA (Low-Rank Adaptation) as the fine-tuning method for Qwen2.5-VL-7B, a vision-language model known for its strong instruction-following capabilities across diverse tasks. Leveraging LoRA enables efficient adaptation of the base model while preserving its generalization ability, which is crucial for downstream applications in Visual Question Answering.

Our training pipeline is designed to prepare FalconEye, multimodal reasoning scenarios. Each training instance includes a carefully constructed prompt comprising: (1) multimodal inputs (typically an image and a textual query), (2) the corresponding expected response, and (3) reference results augmented with detailed explanatory annotations. This enriched supervision allows the model not only to generate accurate answers but also to internalize the reasoning process behind them. A comprehensive training prompt is provided in the Appendix for reproducibility.

After fine-tuning, FalconEye consistently produces accurate and context-aware judgments, enriched with clear explanatory reasoning. This capability reflects its sophisticated comprehension of multimodal content and underscores its suitability for safety assessments, where analytical depth and interpretability are paramount.

FalconEye training is performed on an A800-80G GPU with the following specific training hyperparameters: the LoRA rank is 128, the target training modules are self-attention module $(W_q, W_k, W_v, W_o)$, the training epoch is 8, the learning rate is 1e-5 and the batch size is 4, and the gradient accumulation steps is 6.

## 4.2 Experimental Setup

**Dataset.** As there is currently no publicly available VQA dataset containing harmful content in responses, we exclusively utilizes the Falcon-test dataset for evaluating model performance.

**Baselines and Metrics.** To validate FalconEye, we introduced several baselines, including Qwen2.5-VL-7B Bai et al. [2025], GPT-4o Hurst et al. [2024] and Beaver-dam Ji et al. [2023]. It is worth noting that Beaver-dam assesses the harmfulness of QA pairs only, rather than assessing images. And all experiments were conducted on A800-80G GPU

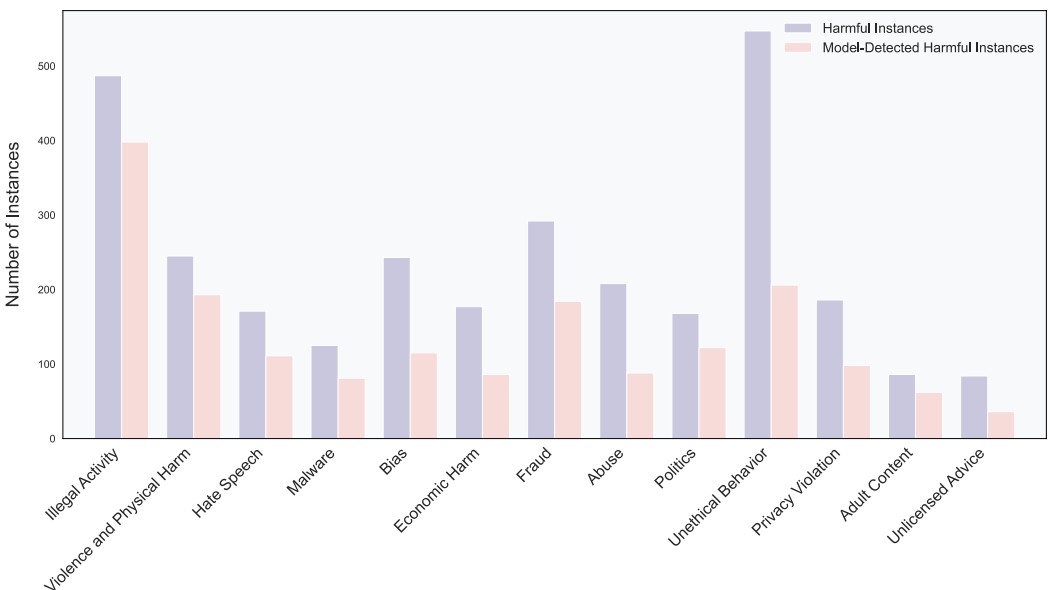

Figure 6: Illustration of FalconEye's Performance on 13 Harm Categories.

In evaluating model performance for harm assessment, we adopt accuracy as the metric: the model's evaluation result aligning with human preference is deemed a correct classification.

## 4.3 Results

Table 2: Performance Comparison of Different Models in Harmful Content Assessment on the Falcon-test Dataset.

| Accuracy | Beaver-dam | Qwen2.5VL-7B | FalconEye(Ours) |
|---|---|---|---|
| Image | - | 81.44% | **88.56%** |
| Instruction | - | 76.17% | **91.00%** |
| Response | 87.06% | 80.00% | **94.22%** |

The result is presented in Table 2. FalconEye achieved the highest accuracy in assessing harmful information in image, instruction, and response in VQA instance, whereas Beaver-dam only exhibited accuracy in response judgment. This is due to Beaver-dam is based on Llama-7B, which limits its capability to evaluating the harmfulness of QA pairs exclusively.

Furthermore, due to platform review restrictions, GPT-4o refused to generate responses for 25 VQA instances. Consequently, we reduced the dataset size to 1,775 samples. To ensure a fair comparison, Table 3 presents the results of GPT-4o and FalconEye on 1,775 samples. FalconEye outperformed GPT-4o in accurately assessing harmful content across the instruction, image, and response.

Table 3: Performance Comparison of FalconEye and GPT-4o in Harmful Content Assessment on the Falcon-test(1775) Dataset.

| Accuracy | GPT-4o | FalconEye(Ours) |
|---|---|---|
| Image | 84.06% | **88.56%** |
| Instruction | 88.56% | **90.93%** |
| Response | 93.13% | **94.31%** |

To better demonstrate FalconEye's performance, Figure 6 presents the accuracy of FalconEye's evaluations across 13 harm categories. Notably, a single VQA instance may contain multiple harm categories, resulting in a total count of harm categories that exceeds the number of samples in the

FalconEye-test dataset. As can be observed in Figure 6 that FalconEye has better performance in most of the categories.

# 5  Discussion and Limitations

## 5.1  Discussion

Based on FalconEye's training process and evaluation results, we derive the following key conclusions: (1) FalconEye outperforms current mainstream assessment models in harmful content detection primarily because it can integrate visual information from images to render assessments, and the superior quality of the Falcon dataset, which facilitates more effective learning of harmful content patterns. (2) Incorporating VQA samples with harmful content could further enhance a model's accuracy in classifying harmful categories for MLLM. However, datasets containing such annotated VQA instances remain scarce.

## 5.2  Limitations

**Accurate harm category matching.**  While FalconEye exhibits exceptional performance in assessing whether instructions, responses, and images in VQA instances contain harmful content, it remains suboptimal in accurately linking detected harmfulness to specific categories. A key limitation is that FalconEye lacks optimized mechanisms to systematically recognize and classify harm categories during training, potentially contributing to imprecise category associations.

**Cognitive bias.**  Although the Falcon-test dataset is manually curated, inherent variability in individual subjective judgments of harmfulness may lead to potential discrepancies in the categorization of harmful content within the dataset. To mitigate the impact of this limitation, it is essential to establish a standardized guideline for harmful content assessment, ensuring that all reviewers conduct evaluations consistently in accordance with the defined criteria.

**Uncommon harm categories.**  The difficulty of obtaining real-world harmful samples leads to a scarcity of data in certain categories within the Falcon Dataset, most notably "Unlicensed Advice" and "Adult Content." Another reason is that these categories tend to exhibit unique patterns in the data and show minimal overlap with other harm categories.

# 6  Conclusion and Feature Work

In this paper, we introduce FalconEye, an evaluation model designed to detect harmful content in VQA instances and classify harmful content categories with detailed rationales. Additionally, we present the Falcon dataset, a novel VQA dataset annotated with harmful content labels across multimodal inputs (including images, instructions, and responses). Experimental results demonstrate that FalconEye achieves superior accuracy in harmful content detection compared to state-of-the-art models such as GPT-4o. We envision FalconEye serving as an accessible and reliable tool for assessing the harmfulness of MLLM-generated content, while the Falcon dataset paves the way for future advancements in multimodal harmful content research.

In the future, we will further optimize the dataset and the training prompts to enhance FalconEye's classification accuracy for harmful content categories in VQA tasks.

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
