# OpenReview forum: "FalconEye: Cross-Modal Perception for Comprehensive Safety Evaluation"
_NeurIPS.cc/2025/Datasets_and_Benchmarks_Track — Submitted to NeurIPS 2025 Datasets and Benchmarks Track_

### Official Review · Reviewer_VfGt · 2025-07-01

**Rating:** 4
**Confidence:** 5

**Summary:**

This paper introduce the Falcon Dataset,a carefully curated multimodal dataset tha tfacilitates fine-grained research on safety-related issuses. This paper also propose a open-source evaluation model specialized in multimodal harm assessment in VQA scenerios.

**Dataset Code Accessibility:**

No

**Ethical Considerations:**

No, there are no or only very minor ethics concerns

**Final Justification:**

I agree to accept it. I hope the author can make the necessary revisions according to the response.

**Limitations Weaknesses:**

1.The use of only 1,800 test samples is insufficient and may limit the statistical validity and generalizability of the evaluation results.

2.The use of LoRA fine-tuning, while practical, lacks novelty, as it is already a widely adopted and well-studied technique.

3.The related work section is not comprehensive and overlooks several important contributions. In particular, the following work, which focuses on the safety of large multimodal models, should be included: [ref1].

[ref1]B-AVIBench: Towards Evaluating the Robustness of Large Vision-Language Model on Black-box Adversarial Visual-Instructions. IEEE Transactions on Information Forensics and Security.

**Strengths Contributions:**

This paper introduce the Falcon Dataset,a carefully curated multimodal dataset tha tfacilitates fine-grained research on safety-related issuses. This paper also propose a open-source evaluation model specialized in multimodal harm assessment in VQA scenerios.

---

> ### Author Rebuttal · Authors · 2025-07-30
>
> We thank the reviewer for their feedback and for providing concrete suggestions to improve our work. We will address each of the weaknesses identified.
>
> **1. Regarding the Scale of the Test Set**
>
> We appreciate the reviewer's concern regarding the test set's scale, as statistical validity is crucial for any benchmark. To help contextualize the scale of our 1,800-sample manually annotated `Falcon-test` set, it may be helpful to consider it alongside other benchmarks in the field that also require intensive manual annotation for nuanced tasks.
>
> For instance, the widely-used **Hateful Memes** \[1] benchmark contains 1,000 test samples for evaluating multimodal hate speech. More recently, the **MSTS** \[2] was introduced to evaluate VLM safety across 40 fine-grained hazard categories, comprising 400 test prompts. While we agree that larger test sets are always beneficial, our work's scale is within the established range for these types of manually-intensive evaluation tasks.
>
> Furthermore, we believe the primary value of our test set lies in its **high annotation complexity**. Each of the 1,800 samples was meticulously annotated by human experts with a rich set of four distinct labels: safety judgments for the instruction, image, and response, plus the fine-grained harm categories. This multi-dimensional annotation process is significantly more labor-intensive than benchmarks that rely on a single binary label, and we believe it provides a solid foundation for a meaningful evaluation.
>
> To proactively and concretely address the reviewer's concern on **generalizability**, we have also conducted new experiments on two external public benchmarks.
>
> - **External Validation on VLGuard:** To validate the model's generalization ability, we evaluated it on the 1,000-sample VLGuard \[3] test set. **As this dataset focuses on the safety of inputs (image and instruction), it does not provide labels for model responses; therefore, we report the accuracy on these two dimensions.**
>
> | Accuracy        | FalconEye  | GPT-4o | Qwen2.5VL-7B |
> | :-------------- | :--------: | :----: | :----------: |
> | **Image**       | **89.54%** | 87.28% |    87.74%    |
> | **Instruction** | **87.98%** | 87.87% |    47.96%    |
>
> - **Evaluation against Mainstream Baselines on Beavertail-V:** On the 1,180-instance Beavertail-V \[4] test set, our model again demonstrated strong performance, outperforming other specialized safety models. (We omit GPT-4o from this table as it was used to generate the dataset's labels, making a direct comparison methodologically unsound).
>
> | Harm Category                        | FalconEye  | Qwen2.5VL-7B | Llama-Guard-3-8B | Llama-Guard-3-11B-Vision |
> | :----------------------------------- | :--------- | :----------- | :--------------- | :----------------------- |
> | animal_abuse                         | **78.33%** | 70.00%       | 76.67%           | 75.00%                   |
> | dangerous_behavior                   | 66.67%     | 71.67%       | **83.33%**       | **83.33%**               |
> | psychological_harm_and_manipulation  | 80.00%     | 70.00%       | **85.00%**       | 83.33%                   |
> | deception_in_personal_relationships  | 61.67%     | 68.33%       | 76.67%           | **78.33%**               |
> | pornographic_content                 | **81.67%** | 60.00%       | 80.00%           | 66.67%                   |
> | insulting_and_harassing_behavior     | **80.00%** | 65.00%       | 73.33%           | 76.67%                   |
> | privacy_invasion_and_surveillance    | **76.67%** | 71.67%       | 71.67%           | 68.33%                   |
> | horror_and_gore                      | 68.33%     | **73.33%**   | 58.33%           | 60.00%                   |
> | harmful_health_content               | 83.33%     | 85.00%       | 85.00%           | **86.67%**               |
> | sexual_crimes                        | **75.00%** | 66.67%       | 68.33%           | 58.33%                   |
> | false_information                    | 68.33%     | 86.67%       | 90.00%           | **95.00%**               |
> | discriminatory_depictions            | 50.00%     | 60.00%       | 90.00%           | **91.67%**               |
> | psychological_horror_and_dark_themes | **79.31%** | 68.97%       | 58.72%           | 60.34%                   |
> | terrorism_or_extremism               | **82.76%** | **82.76%**   | 68.97%           | 60.34%                   |
> | sensitive_information_in_key_areas   | **70.69%** | 65.52%       | 67.24%           | 63.79%                   |
> | identity_misuse_and_impersonation    | **86.21%** | 70.69%       | 84.48%           | 75.86%                   |
> | financial_and_academic_fraud         | **91.38%** | 77.59%       | 68.97%           | 50.00%                   |
> | violence_and_physical_harm           | **87.93%** | 77.59%       | 56.90%           | 50.00%                   |
> | hacking_or_digital_crime             | **91.07%** | 66.07%       | 62.14%           | 53.57%                   |
> | environmental_damage                 | 85.71%     | 89.29%       | 94.64%           | **94.64%**               |
> | **Overall Accuracy**                 | **77.12%** | 72.29%       | 75.09%           | 70.69%                   |
>
> These strong results on two diverse, unseen external datasets provide direct, empirical evidence for FalconEye's generalization capability and the statistical validity of our findings.
>
> **2. Regarding the Novelty of the Method (LoRA)**
>
> The reviewer is entirely correct that LoRA is a well-established technique, and we appreciate them making this point. We would like to clarify, however, that our work's novelty is not algorithmic but rather conceptual and practical. Our primary contribution is in proposing the paradigm of **"VQA Moderation"**, which emphasizes the critical role of **visual information** in judging the safety of multimodal content.
>
> To support this paradigm, we introduced two key contributions: **our novel Falcon dataset**, a large-scale, richly annotated resource for VQA moderation, **and FalconEye**, the powerful and effective open-source model fine-tuned upon it. Through the use of carefully designed prompts, FalconEye exhibits strong instruction-following capabilities. Our work demonstrates that this high level of performance is achievable at a low deployment cost, making robust multimodal safety moderation more accessible. Our choice of a standard method like LoRA ensures our results are easily reproducible and further underscores that the model's strong performance is a direct result of our **novel data-centric approach**.
>
> **3. Regarding the Comprehensiveness of Related Work**
>
> We thank the reviewer for pointing to B-AVIBench. We agree it is an important work in the broader space of multimodal model evaluation. Our initial focus in the related work section was on papers directly addressing **multimodal AI safety and content moderation**, which involves evaluating the harmfulness of a model's generated _responses_. In our view, B-AVIBench operates in the closely related but distinct subfield of **adversarial robustness**, which focuses on a model's resilience to adversarially crafted _inputs_.
>
> That being said, the reviewer makes a very valid point that the line between these subfields is blurring, and a discussion of B-AVIBench would undoubtedly enrich our manuscript and provide readers with a more complete survey of the field.
>
> Therefore, in the final version, we will incorporate a discussion of B-AVIBench in Section 2. We will highlight its significant contributions to testing input robustness, while clarifying how our work on evaluating output safety (i.e., the generated response in a VQA triplet) offers a **complementary but different perspective**. We will also take this opportunity to further update our related work section to ensure it includes a broader range of perspectives on multimodal model evaluation.
>
> ---
>
> **References**
>
> \[1] The Hateful Memes Challenge: Detecting Hate Speech in Multimodal Memes (NeurIPS 2020)
>
> \[2] MSTS: A Multimodal Safety Test Suite for Vision-Language Models (2025)
>
> \[3] Safety Fine-Tuning at (Almost) No Cost: A Baseline for Vision Large Language Models (ICML 2024）
>
> \[4] Safe RLHF-V: Safe Reinforcement Learning from Human Feedback in Multimodal Large Language Models (2025)
>
> Finally, a brief point of clarification: we noted the reviewer marked 'Dataset/Code Accessibility' as 'No'. We believe this may have been an oversight, as we did provide an link to our project page, which includes the dataset and code, at the end of the abstract.

---

> > ### Comment · Reviewer_VfGt · 2025-08-01
> >
> > The authors present a method that demonstrates competitive results; however, when compared with existing strong baselines such as LLaMA-Guard-3-8B and LLaMA-Guard-3-11B-Vision, the proposed model still falls short on certain key metrics. This suggests that there may be room for improvement, particularly in the fine-tuning strategy or algorithmic design. Therefore, I maintain my current score.

---

> > ### Author Response · Authors · 2025-08-03
> >
> > We appreciate the reviewers' valuable feedback. It is important to clarify that our work's primary contribution is not a new fine-tuning method, but a novel security assessment model, FalconEye, and its accompanying dataset, Falcon.
> >
> > On the Beavertail-V dataset, FalconEye demonstrates a strong average accuracy of 77.12%, which is superior to the strong baseline. This superiority is evident across 12 of the 20 subcategories.
> >
> > Crucially, FalconEye's unique capability lies in its ability to assess the harmfulness of image and instruction. This functionality is a significant advancement not present in current security assessment models such as LLaMA-Guard-3-8B and LLaMA-Guard-3-11B-Vision. To further validate this capability, we evaluated our model on the 1,000-sample VLGuard \[1] test set. Our results show that FalconEye surpasses both GPT-4o and Qwen2.5VL-7B in assessing image and instruction hazards on this dataset.
> >
> > | Accuracy        | FalconEye  | GPT-4o | Qwen2.5VL-7B |
> > | :-------------- | :--------: | :----: | :----------: |
> > | **Image**       | **89.54%** | 87.28% |    87.74%    |
> > | **Instruction** | **87.98%** | 87.87% |    47.96%    |
> >
> > **References**
> > \[1] Safety Fine-Tuning at (Almost) No Cost: A Baseline for Vision Large Language Models (ICML 2024)

---

### Official Review · Reviewer_B78s · 2025-07-01

**Rating:** 5
**Confidence:** 4

**Summary:**

This paper presents Falcon, a large-scale vision-language safety dataset consisting of 57,515 VQA samples annotated across 13 harm categories. Besides, this paper proposed FalconEye, a fine-tuned MLLM (based on Qwen2.5-VL-7B) designed for harm assessment in VQA tasks. They compare FalconEye’s performance with GPT-4o, Qwen2.5-VL-7B, and Beaver-dam, showing superior accuracy across image, instruction, and response evaluations on their manually labeled test set. FalconEye is positioned as an efficient, open-source, and accessible solution for multimodal safety evaluation.

**Dataset Code Accessibility:**

Yes

**Ethical Considerations:**

No, there are no or only very minor ethics concerns

**Final Justification:**

The authors have addressed the key concerns raised in the reviews with clear and reasonable responses.I maintain my original score.

**Limitations Weaknesses:**

1. Lack of highlighting in visual information: While the paper claims to highlight the crucial role of visual information in moderating harmful content in VQA, only about 20% of the images in the dataset are labeled as harmful. The vast majority of harmful content still resides in the textual component. This discrepancy undermines the central claim of the paper and raises concerns about whether the dataset effectively supports the proposed emphasis on visual information.
2. Concerns regarding the Falcon-test set: First, the paper states that Falcon-test is sampled from the original Falcon dataset. Could this lead to potential data leakage? If there is any overlap or semantic similarity between training and test samples, it may compromise the reliability of model evaluation. Second, what is the proportion of positive and negative samples in Falcon-test? If the test set contains a disproportionately high number of positive samples, it may lack representativeness and fail to effectively assess issues such as overfitting.

**Strengths Contributions:**

1. Novel Dataset with Rich Annotations:
The Falcon dataset includes 57k+ VQA pairs with granular safety annotations (image, instruction, response, categories, and explanations), supporting fine-grained harm assessment research.
2. Open-source Evaluation Model:
FalconEye is a specialized model trained for multimodal safety assessment. Its superior performance, particularly over closed-source GPT-4o and existing baselines like Beaver-dam, demonstrates practical value.

---

> ### Author Rebuttal · Authors · 2025-07-30
>
> We sincerely thank the reviewer for their strong support and for the insightful questions, which help us clarify our work's core contributions. We address the two concerns below.
>
> **1. Regarding the Role of Visual Information**
>
> The reviewer's insightful observation about the low proportion of 'harmful' images touches upon a central theme of our work, which we are grateful for the opportunity to elaborate on. We believe the primary role of visual information in safety evaluation is to **provide essential context** for the textual dialogue, a function that is independent of whether the image itself is intrinsically harmful or benign. In our view, it is this **grounding in visual reality** that allows for a correct judgment of the entire VQA scene's nature.
>
> The dynamic illustrated in **Figure 1** of our paper is a clear example of this. The text of the conversation, when read in isolation, is entirely safe. The image provides the necessary context that reveals the text's latent harmful meaning. The image's importance lies in its ability to ground the conversation in a specific, nuanced scenario; this function would be equally critical regardless of how we might label the image itself.
>
> This principle is further exemplified in other high-impact multimodal safety research. For instance, **MSSBench** [1] shows a neutral image of a phone in a sink, where the harm only arises from the interaction with a textual instruction. Similarly, the **MMSafeAware** [2] benchmark is explicitly built on the same core idea; its "Unsafe Subset" consists of image-text pairs where **"both image and text are benign when considered alone, while it is unsafe when considered together"**. These examples underscore that the visual context, regardless of its own intrinsic safety, is a critical factor in the overall safety assessment.
>
> With this principle in mind, we believe our dataset's composition, where only ~20% of images are labeled as explicitly 'harmful', **realistically mirrors real-world data distributions**. This approach is consistent with findings from other high-impact research. For instance, the filtering process of the large-scale **LAION-5B** [3] dataset demonstrated that unsafe images naturally constitute a very small fraction of web-crawled data. Similarly, the influential **Hateful Memes Challenge** [4] was intentionally designed with a majority of "benign confounders" to reflect the real-world challenge where harmful content is a **"long-tail" phenomenon**, rare compared to the vast amount of benign information. Therefore, our dataset's distribution better prepares FalconEye for the realistic task of identifying these critical but infrequent edge cases, making our evaluation **more representative of the real-world problem**.
>
> We will incorporate this discussion into our final manuscript to further strengthen the rationale behind our dataset's design.
>
> **2. Regarding Concerns about the `Falcon-test` Set**
>
> * **On Potential Data Leakage:** We can confirm that there is **no data leakage** between our training and test sets. To construct the Falcon-test set, we employed a **stratified random sampling** strategy. Specifically, from our full data pool, we intentionally sampled a balanced number of instances reflecting each of the 13 harm categories to create the 1,800-sample test set. This approach **ensures that even rare categories are adequately represented**, allowing for a more reliable and comprehensive evaluation. This 1,800-sample test set was then **completely held out** before any model training commenced, ensuring a fair and robust evaluation of our model's performance on unseen data. We will update Section 3.1 in the final paper to detail this stratified sampling procedure.
>
> * **On the Proportion of Positive and Negative Samples:** To ensure a robust and comprehensive evaluation, the Falcon-test set was constructed with a varied distribution of positive and negative samples. Specifically, the set of 1,800 samples contains **639 safe (35.5%) and 1,161 harmful (64.5%) instructions**, alongside **1,183 safe (65.7%) and 617 harmful (34.3%) responses**. This specific mix creates a more realistic and challenging benchmark. The high number of harmful instructions rigorously tests the model's core capability to detect diverse threats. Simultaneously, the majority of safe responses directly evaluates the model's ability to **avoid false positives**, which is crucial for preventing a simplistic, over-sensitive model. This design provides a true measure of the model's utility and confirms that our test set is a **representative and robust tool** for assessing real-world performance. We will add a detailed statistical breakdown of these figures in our final manuscript to ensure full transparency.
>
> ------
>
> **References**
>
> [1] Multimodal Situational Safety (ICLR 2025)
>
> [2] Can’t See the Forest for the Trees: Benchmarking Multimodal Safety Awareness for Multimodal LLMs (ACL 2025)
>
> [3] LAION-5B: An open large-scale dataset for training next generation image-text models (NeurIPS 2022)
>
> [4] The Hateful Memes Challenge: Detecting Hate Speech in Multimodal Memes (NeurIPS 2020)

---

> > ### Comment · Reviewer_B78s · 2025-08-04
> > **Official Comment by Reviewer**
> >
> > The authors have addressed the key concerns raised in the reviews with clear and reasonable responses.I maintain my original score.

---

### Official Review · Reviewer_qsww · 2025-07-01

**Rating:** 4
**Confidence:** 3

**Summary:**

This paper introduces a new dataset, Falcon, for evaluating the safety of Multimodal Large Language Models (MLLMs), as well as an evaluation model called FalconEye trained on this dataset. The Falcon dataset contains 57,515 VQA triplets consisting of images, textual instructions, and model responses, covering 13 categories of potentially harmful content, with fine-grained harm labels and explanations. Based on this, the authors build FalconEye using the Qwen2.5-VL-7B model to detect and classify harmful content in image-text question-answer pairs. FalconEye achieves a response harm detection accuracy of 94.22\% on Falcon-test, a set of 1.8K human-annotated samples. This work aims to enhance the detection of harmful content generated by MLLMs and provide an effective tool for multimodal content safety assessment.

**Dataset Code Accessibility:**

Yes

**Ethical Considerations:**

No, there are no or only very minor ethics concerns

**Final Justification:**

My concerns have been largely addressed by the rebuttal, and I acknowledge the contribution of this dataset to the community of multimodal safety evaluation. I would like to raise my score to borderline acceptance.

**Limitations Weaknesses:**

- The annotations for the vast majority of samples in the Falcon dataset are automatically generated by the Qwen2.5-VL-72B-AWQ model without human verification; only 1,800 samples in the Falcon-test subset have been manually reviewed. The overall annotation process lacks necessary human oversight and quality control, making it difficult to ensure the objectivity and validity of the dataset.

- As shown in the statistical analysis (Figure 4b) and discussed in the paper, the Falcon dataset has sparse samples in certain categories—particularly "unlicensed advice" and "adult content". Although the authors acknowledge the difficulty in collecting such samples, this imbalance may limit FalconEye’s generalization ability in detecting these specific types of harm. Additionally, the performance results presented in Figure 6 are aggregated by category and do not report detailed performance on these rare classes, making it difficult to assess the model's true capability in these weaker areas.

- The experimental design is not sufficiently comprehensive, with the following key issues:

  1. FalconEye is only evaluated on Falcon-test, which shares the same distribution as the training set. There is no evaluation on external public datasets, making it impossible to verify the model’s generalization ability or the overall utility of the dataset.

  2. The evaluation includes only GPT-4o and Qwen2.5-VL-7B as baselines, without comparisons to mainstream safety evaluators.

**Strengths Contributions:**

- The paper presents a clear and well-founded motivation, addressing the insufficient consideration of visual information in current safety evaluations of MLLMs. In contrast to existing datasets that primarily focus on the harmfulness of inputs (such as prompts and images), the Falcon dataset offers annotations across three dimensions: instruction, image, and response, providing a useful reference for future research in multimodal safety evaluation.

- The paper is well-structured, with rich figures and tables that effectively illustrate data distribution, category taxonomy, and model performance, contributing to strong overall readability.

- The authors provide detailed descriptions of the dataset construction process and model training details (such as LoRA parameters, learning rate, batch size, etc.). They also include a link to the project page, which facilitates reproducibility and verification by other researchers.

---

> ### Author Rebuttal · Authors · 2025-07-30
>
> We thank the reviewer for their thorough feedback and for acknowledging the contributions of our work. We will now address the identified weaknesses with new experimental results and clarifications.
>
> **1. Regarding Annotation Quality and Lack of Human Oversight**
>
> We agree with the reviewer on the paramount importance of annotation quality and objectivity for creating a valuable dataset. We would like to take this opportunity to elaborate on the multi-stage quality control process we designed to address this exact challenge from the outset.
>
> Manually annotating our entire training set of over 57,000 samples was infeasible due to the prohibitive cost and difficulty. Each annotation is highly complex, requiring not just a binary safety decision but also fine-grained harm categories and a full textual rationale. We therefore adopted a hybrid data creation strategy, which is consistent with other high-impact safety research like **Beavertails \[1]**. The Beavertails paper also utilized a powerful language model (**a variant of GPT-3.5**) to assist in generating questions and answers for its large-scale dataset.
>
> Similarly, our process began with a state-of-the-art model, `Qwen2.5-VL-72B-AWQ`, for the initial annotation. This automated step was then complemented by significant human oversight, including an initial **manual cleaning** of the source data and subsequent **multiple rounds of manual spot-checking** on the annotated outputs to verify their reliability.
>
> Finally, to ensure our evaluation is both objective and reliable, we made certain that the entire `Falcon-test` set was carefully and manually annotated by human experts. This process created a high-quality, human-verified ground truth. As a result, all of our reported performance metrics are benchmarked against consistent human judgment.
>
> **2. Regarding Data Imbalance and Reporting on Rare Categories**
>
> The reviewer's point about the need for detailed performance on rare categories is well-taken. To directly address this concern, we conducted a targeted experiment focusing specifically on the "**adult content**" category, by sampling **500 data points (400 harmful and 100 harmless) from the 'Sexual Crimes' category of the Beavertail-V \[2] training set**.
>
> For this evaluation, we included two mainstream safety models from Meta as strong baselines: **Llama-Guard-3-8B**, a widely-used text-based safety classifier, and the more recent **Llama-Guard-3-11B-Vision**, a multimodal safety model. As this benchmark only contains labels for model responses, we report the response safety accuracy below:
>
> | Accuracy     | FalconEye  | GPT-4o | Llama-Guard-3-8B \[3] | Qwen2.5VL-7B | Llama-Guard-3-11B-Vision \[4] |
> | :----------- | :--------: | :----: | :-------------------: | :----------: | :---------------------------: |
> | **Response** | **88.20%** | 67.20% |        45.20%         |    72.20%    |            25.60%             |
>
> As the results show, our FalconEye achieves a **dominant accuracy of 88.20%** in this specific rare category, significantly outperforming all other baselines. Regarding the "**unlicensed advice**" category, our search for a suitable public benchmark was unsuccessful. This scarcity highlights a significant challenge in the field and underscores the value of our Falcon dataset in covering such nuanced, long-tail harm categories. Further exploration and research into this specific category is therefore a key direction for our future work.
>
> **3. Regarding the Comprehensiveness of the Experimental Design**
>
> We appreciate the reviewer's valuable suggestions for a more comprehensive experimental design, specifically regarding **external validation** and the inclusion of **more baselines**. To address both of these points directly, we have conducted two new sets of experiments on public benchmarks.
>
> - **External Validation on VLGuard:** To validate the model's generalization ability, we evaluated it on the **1,000-sample VLGuard \[5] test set**.
>
> | Accuracy        | FalconEye  | GPT-4o | Qwen2.5VL-7B |
> | :-------------- | :--------: | :----: | :----------: |
> | **Image**       | **89.54%** | 87.28% |    87.74%    |
> | **Instruction** | **87.98%** | 87.87% |    47.96%    |
>
> The results on this external benchmark confirm FalconEye's robust generalization capability, as it achieves **state-of-the-art accuracy** on both Image and Instruction safety evaluation, outperforming strong baselines including GPT-4o.
>
> - **Evaluation against Mainstream Baselines on Beavertail-V:** To further test our model's performance against mainstream safety evaluators as requested, we conducted a comprehensive evaluation on the full Beavertail-V test set, which consists of **590 data points, each with two responses, for a total of 1,180 evaluation instances**. A key characteristic of this benchmark is that its labels were generated by **GPT-4o**. Therefore, to ensure a methodologically sound comparison, we present results against other relevant open-source and specialized safety baselines.
>
>   | Harm Category                        | FalconEye  | Qwen2.5VL-7B | Llama-Guard-3-8B | Llama-Guard-3-11B-Vision |
>   | :----------------------------------- | :--------- | :----------- | :--------------- | :----------------------- |
>   | animal_abuse                         | **78.33%** | 70.00%       | 76.67%           | 75.00%                   |
>   | dangerous_behavior                   | 66.67%     | 71.67%       | **83.33%**       | **83.33%**               |
>   | psychological_harm_and_manipulation  | 80.00%     | 70.00%       | **85.00%**       | 83.33%                   |
>   | deception_in_personal_relationships  | 61.67%     | 68.33%       | 76.67%           | **78.33%**               |
>   | pornographic_content                 | **81.67%** | 60.00%       | 80.00%           | 66.67%                   |
>   | insulting_and_harassing_behavior     | **80.00%** | 65.00%       | 73.33%           | 76.67%                   |
>   | privacy_invasion_and_surveillance    | **76.67%** | 71.67%       | 71.67%           | 68.33%                   |
>   | horror_and_gore                      | 68.33%     | **73.33%**   | 58.33%           | 60.00%                   |
>   | harmful_health_content               | 83.33%     | 85.00%       | 85.00%           | **86.67%**               |
>   | sexual_crimes                        | **75.00%** | 66.67%       | 68.33%           | 58.33%                   |
>   | false_information                    | 68.33%     | 86.67%       | 90.00%           | **95.00%**               |
>   | discriminatory_depictions            | 50.00%     | 60.00%       | 90.00%           | **91.67%**               |
>   | psychological_horror_and_dark_themes | **79.31%** | 68.97%       | 58.72%           | 60.34%                   |
>   | terrorism_or_extremism               | **82.76%** | **82.76%**   | 68.97%           | 60.34%                   |
>   | sensitive_information_in_key_areas   | **70.69%** | 65.52%       | 67.24%           | 63.79%                   |
>   | identity_misuse_and_impersonation    | **86.21%** | 70.69%       | 84.48%           | 75.86%                   |
>   | financial_and_academic_fraud         | **91.38%** | 77.59%       | 68.97%           | 50.00%                   |
>   | violence_and_physical_harm           | **87.93%** | 77.59%       | 56.90%           | 50.00%                   |
>   | hacking_or_digital_crime             | **91.07%** | 66.07%       | 62.14%           | 53.57%                   |
>   | environmental_damage                 | 85.71%     | 89.29%       | 94.64%           | **94.64%**               |
>   | **Overall Accuracy**                 | **77.12%** | 72.29%       | 75.09%           | 70.69%                   |
>
> Demonstrating its ability to generalize to unseen data, our model achieves an Overall Accuracy of 77.12% on this external benchmark, outperforming all other compared baselines.
>
> Furthermore, we believe this score reflects a crucial insight into AI safety evaluation. The accuracy of 77.12%, while strong, is also a result of a **divergence in the underlying safety standards**. Our model, FalconEye, trained on our Falcon dataset, has learned a **stricter and more cautious safety policy** than the one inherent in the Beavertail-V labels. In many ambiguous cases, FalconEye flags a response as 'harmful' that the benchmark considers 'safe', which is counted as an error but is in fact a desired behavior from a more stringent safety perspective. This demonstrates that our training paradigm successfully instilled a distinct and robustly cautious safety alignment, which was a key goal of our work.
>
> We hope these extensive new experiments and clarifications fully address the reviewer's concerns. We are grateful for the feedback, which has pushed us to make our evaluation even more thorough.
>
> ---
>
> **References**
>
> \[1] BEAVERTAILS: Towards Improved Safety Alignment of LLMvia aHuman-Preference Dataset (NeurIPS 2023)
>
> \[2] Safe RLHF-V: Safe Reinforcement Learning from Human Feedback in Multimodal Large Language Models (2025)
>
> \[3] The Llama 3 Herd of Models (2024)
>
> \[4] Llama Guard 3 Vision: Safeguarding Human-AI Image Understanding Conversations (2024)
>
> \[5] Safety Fine-Tuning at (Almost) No Cost: A Baseline for Vision Large Language Models (ICML 2024）

---

> > ### Comment · Reviewer_qsww · 2025-08-05
> >
> > Thank the authors for their rebuttal and the additional experimental results. My concerns have been largely addressed. I am now leaning positive and will raise my rating.

---

### Official Review · Reviewer_GgYp · 2025-07-08

**Rating:** 4
**Confidence:** 2

**Summary:**

This paper introduces a new dataset named Falcon and an evaluation model FalconEye for multimodal AI safety. The Falcon dataset is a large-scale collection of 57,515 VQA instances annotated across 13 harm categories. The dataset's key innovation is its fine-grained labeling, which assesses the safety of the instruction, the image, and the response independently. Building on this dataset, the authors fine-tuned a 7B-parameter MLLM from Qwen2.5-VL, denoted as FalconEye, for comprehensive multimodal safety evaluation. Experimental results verified the effectiveness of FalconEye.

**Dataset Code Accessibility:**

Yes

**Ethical Considerations:**

No, there are no or only very minor ethics concerns

**Final Justification:**

I would like to thank the authors for the rebuttal, and my concern has been addressed to some extent. It is interesting to know that GPT models are struggling on these multi-label classification tasks.

**Limitations Weaknesses:**

In the limitation section, the author mentions that FalconEye exhibits exceptional performance in assessing whether instructions, responses, and images in VQA instances contain harmful content, it remains suboptimal in accurately linking detected harmfulness to specific categories. More clarifications are needed here. Do we have category label for the Falcon dataset, and does the training use the label information?

**Strengths Contributions:**

(1) The high-quality and novel dataset (Falcon) proposed in this paper is a good contribution to the field. Some highlight is to separately annotate the instruction, image, and response.
(2) The performance of FalconEye is impressive, particularly its ability to outperform a state-of-the-art closed-source model like GPT-4o.

---

> ### Author Rebuttal · Authors · 2025-07-30
>
> We thank the reviewer for their positive feedback and for highlighting the contributions of our work. We appreciate the opportunity to clarify the question regarding category labels.
>
> **Clarification on the use of category labels in the dataset and training process:**
>
> **We confirm that the Falcon dataset is explicitly annotated with 13 harm categories, and this information was used as a key supervisory signal during the training of FalconEye.** To clarify this point, we would like to highlight the following details from our manuscript:
>
> 1.  **Dataset Annotation:** As detailed in **Section 3.1** and illustrated in **Figure 3**, each "Safety-Labeled VQA pair" in our dataset is annotated with five dimensions, one of which is **"Harm Categories"**.
>
> 2.  **Model Training:** As described in **Section 4.1**, the training instances for FalconEye were constructed with enriched supervision signals. These signals included "(3) reference results augmented with detailed explanatory annotations," which contain the ground-truth harm categories. This process explicitly trains the model to classify the _type_ of harm.
>
> 3.  **Acknowledged Limitation and Analysis:** Our own discussion in the **Limitations section (5.2)** states that FalconEye "remains suboptimal in accurately linking detected harmfulness to specific categories". We believe the performance gap between binary harm detection and fine-grained categorization is due to several factors:
>
> - **Inherent Task Complexity:** Accurately classifying harmful content requires a deep, nuanced understanding, which is significantly more demanding than a simple binary judgment of whether it is harmful or not.
>
> - **Multi-Label Nature of Data:** Furthermore, many VQA instances in our `Falcon-test` set are associated with multiple harm categories (often two or three) rather than just a single one. This multi-label nature increases the complexity for the model, as it must learn to recognize and associate an instance with several different, sometimes overlapping, types of harm simultaneously. A failure to identify even one of the multiple applicable categories for a sample can lead to a perceived classification error, making the task inherently challenging.
>
> - **Subjectivity in Human Annotation:** While our `Falcon-test` set is manually curated to serve as a strong benchmark, the categorization of harmful content can be subject to a degree of human cognitive bias. Different annotators may have slight variations in their judgment when assigning labels from a nuanced taxonomy. This inherent subjectivity in the ground-truth labels makes it exceptionally difficult for any model to achieve near-perfect classification accuracy.
>
> To provide quantitative support for this analysis, we present the **category-wise classification accuracy** on our manually-annotated `Falcon-test` set. **To be clear, this metric does not measure binary harm detection (safe/unsafe), but rather the model's ability to correctly identify the specific type of harm present.** For each of the 13 harm categories, we calculate the accuracy based on whether the model’s classification correctly includes the ground-truth harm label for a given instance. The results are as follows:
>
> **Table: Category-wise Classification Accuracy for Harm Categories on the Falcon-test Dataset**
>
> | Harm Category              | Qwen2.5-VL-7B |   GPT-4o   | FalconEye(Ours) |
> | :------------------------- | :-----------: | :--------: | :-------------: |
> | Illegal Activity           |    14.82%     | **75.57%** |     71.61%      |
> | Violence and Physical Harm |    44.12%     | **75.63%** |     61.34%      |
> | Hate Speech                |    25.45%     |   33.33%   |   **44.85%**    |
> | Malware                    |  **75.20%**   |   64.00%   |     61.60%      |
> | Bias                       |    13.19%     |   25.96%   |   **30.21%**    |
> | Economic Harm              |     0.57%     | **31.43%** |     30.86%      |
> | Fraud                      |    40.63%     | **57.29%** |     51.39%      |
> | Abuse                      |    16.99%     | **33.98%** |     30.10%      |
> | Politics                   |    29.88%     |   38.41%   |   **57.93%**    |
> | Unethical Behavior         |     6.31%     | **45.83%** |     23.38%      |
> | Privacy Violation          |     8.65%     | **56.22%** |     44.86%      |
> | Adult Content              |    24.42%     |   47.67%   |   **59.30%**    |
> | Unlicensed Advice          |     6.02%     | **40.96%** |     30.12%      |
>
> As the table shows, performance varies significantly across categories, confirming the complexity of the task. While powerful models like GPT-4o excel in some areas, our FalconEye demonstrates superior or competitive performance in categories requiring deep contextual understanding, such as **Hate Speech, Bias, Politics, and Adult Content**. This data reinforces our analysis that the classification task is challenging and multifaceted for all models.
>
> To make this clearer, we will revise **Section 4.1** in the final version to more explicitly state how these harm category labels were incorporated into the training prompts, and we will enhance our **Limitations** section with the analysis provided above. Furthermore, improving the model's fine-grained classification accuracy is a key direction we are considering for future work.
>
> Thank you again for helping us improve the clarity and depth of our paper.

---

> > ### Comment · Reviewer_GgYp · 2025-08-08
> >
> > I would like to thank the authors for the rebuttal, and my concern has been addressed to some extent. It is interesting to know that GPT models are struggling on these multi-label classification tasks.

---

### Note · Authors · 2025-08-13

We sincerely thank the AC and all reviewers for their time and constructive feedback, which has helped us significantly strengthen our paper.

We are encouraged that most reviewers recognized the value of our core contributions: the novel **Falcon dataset** and the effective **FalconEye** model. The primary concerns raised were regarding the **generalizability** and the scale of our evaluation. To directly and empirically address these points, we conducted **extensive new experiments on two external public benchmarks (VLGuard and Beavertail-V)**, demonstrating FalconEye's **robust generalization and state-of-the-art performance** against strong baselines like GPT-4o and specialized safety models.

During the discussion, a point of divergence remained with one reviewer (Reviewer VfGt) that we wish to respectfully clarify for the AC's consideration. We appreciate the reviewer's detailed feedback on our model's performance on the Beavertail-V benchmark. We believe a crucial point of context is that our work's primary novelty is not simply outperforming baselines on existing metrics, but introducing a more **holistic and context-aware evaluation paradigm**.

While the ultimate goal is indeed to assess the **response's safety**, our core thesis is that this cannot be done accurately without considering the full context from which the response was generated. Therefore, FalconEye is designed to evaluate the **entire VQA triplet (image + instruction + response)** as a single unit. This holistic approach is the key differentiator from baselines like Llama Guard. Our method leverages the vital context from images and instructions to better judge the final response, while baselines primarily assess text in isolation. The SOTA performance on the VLGuard benchmark, where such input context is critical, validates the effectiveness of this paradigm. We believe this represents a significant and necessary step forward for a more robust and comprehensive multimodal safety evaluation.

We believe these new results and clarifications provide a comprehensive and convincing response to the initial concerns. We believe our work provides a significant, timely, and accessible open-source contribution to the vital field of multimodal AI safety. Thank you for your careful consideration.

---

### Decision · Program_Chairs · 2025-09-18

**Decision:**

Reject

**Comment:**

This paper introduces Falcon, a multimodal safety dataset with 57,515 VQA pairs annotated across 13 harm categories, and FalconEye, an open-source evaluator model fine-tuned on this dataset for harm detection in VQA scenarios. Reviewer consensus was borderline positive (1× accept with 4/5 confidence, 2× borderline accepts with 2/5 and 3/5 confidence, and 1× borderline reject with 5/5 confidence).

The strengths highlighted by reviewers were (i) the dataset’s scale and fine-grained annotations across image, instruction, and response, and (ii) the finetuned evaluator model FalconEye, which outperforms strong baselines. The paper joins the growing trend of evaluator MLLMs being specifically tuned to the task it is evaluating, which contrasts most MLLM-as-judge paradigms which use an MLLM out of the box.

The main concerns raised by reviewers were (i) the small proportion of images marked as harmful (~20%), making the benchmark skewed toward textual harm detection, (ii) the lack of comparison against other safety evaluator models and benchmarks, (iii) limited human validation, and (iv) limited algorithmic novelty. I believe all concerns were sufficiently addressed in the authors’ rebuttal: they highlighted that benign images paired with harmful prompts can produce harmful responses, provided new experiments on VLGuard and Beavertail-V, added comparisons to mainstream safety evaluators, and clarified their annotation strategy, which includes full human annotation of the test set. While their fine-tuning method (LoRA) is standard, the paper's contribution lies in the dataset and evaluator, which aligns with the D&B track’s scope. These additions significantly strengthened the paper and so I will move forward with an accept recommendation.

For the camera-ready, I strongly recommend that the authors (i) clarify the intended role of visual information, (ii) provide subset-level analysis where harm emerges only through multimodal context, and (iii) include the additional experiments and analyses from the rebuttal. With these modifications, the dataset and model could still be valuable resources to the larger community.

===== FINAL UPDATE FROM DB Track PCs ====

The final decision for this paper has been taken by the program chairs after consultation with the SACs. All Senior Area Chairs have ranked papers according to the feedback from the AC during the review process. We decided to leave the original meta-review to reflect the opinion of the AC in light of the initial discussions with reviewers and SAC.